# Phyto-Assisted Synthesis of Nanoselenium–Surface Modification and Stabilization by Polyphenols and Pectins Derived from Agricultural Wastes

**DOI:** 10.3390/foods12051117

**Published:** 2023-03-06

**Authors:** Nikolina Golub, Emerik Galić, Kristina Radić, Ana-Maria Jagodić, Nela Predović, Kristina Katelan, Lucija Tesla, Sandra Pedisić, Tomislav Vinković, Dubravka Vitali Čepo

**Affiliations:** 1Faculty of Pharmacy and Biochemistry, University of Zagreb, 10000 Zagreb, Croatia; 2Faculty of Agrobiotechnical Sciences Osijek, Josip Juraj Strossmayer University of Osijek, 31000 Osijek, Croatia; 3Faculty of Food Technology and Biotechnology, University of Zagreb, 10000 Zagreb, Croatia

**Keywords:** selenium nanoparticles, pectin, olive pomace extract, green synthesis, antioxidant activity

## Abstract

Raw and purified mandarin peel-derived pectins were characterized and combined with olive pomace extract (OPE) in the green synthesis of selenium nanoparticles (SeNPs). SeNPs were characterized in terms of size distribution and zeta potential, and their stability was monitored during 30 days of storage. HepG2 and Caco-2 cell models were used for the assessment of biocompatibility, while antioxidant activity was investigated by the combination of chemical and cellular-based assays. SeNP average diameters ranged from 171.3 nm up to 216.9 nm; smaller SeNPs were obtained by the utilization of purified pectins, and functionalization with OPE slightly increased the average. At concentrations of 15 mg/L SeNPs were found to be biocompatible, and their toxicity was significantly lower in comparison to inorganic selenium forms. Functionalization of SeNPs with OPE increased their antioxidant activity in chemical models. The effect was not clear in cell-based models, even though all investigated SeNPs improved cell viability and protected intracellular reduced GSH under induced oxidative stress conditions in both investigated cell lines. Exposure of cell lines to SeNPs did not prevent ROS formation after exposure to prooxidant, probably due to low transepithelial permeability. Future studies should focus on further improving the bioavailability/permeability of SeNPs and enhancing the utilization of easily available secondary raw materials in the process of phyto-mediated SeNP synthesis.

## 1. Introduction

Selenium (Se) is an essential metalloid involved in different physiological functions that have been attributed largely to its presence in selenoproteins in the form of the 21st amino acid, selenocysteine. Se modulates a wide spectrum of biological processes: redox signaling, cellular differentiation, immune response, cellular response to oxidative stress and protein folding [1]. Its intake is extremely variable across the world due to significant differences in the content and availability of the soil and can result in both Se deficiency and excessive intake. Se deficiency affects approximately 1 billion people worldwide. Usually, it occurs in areas where soil selenium content is poor as well as in certain pathological conditions (patients receiving parenteral nutrition, cirrhosis patients due to ineffective selenomethionine metabolism, low-birth-weight infants etc.). It has been implicated in the pathogenesis of cardiovascular disease, infertility, myodegenerative diseases and cognitive decline [2]. In food and dietary supplements, Se exists in several forms, including selenide (Se^2−^), selenite (SeO_3_^2−^), selenate (SeO_4_^2−^), selenocysteine (Se-Cys), selenometheonine (Se-Met) and elemental nanoselenium [3,4].

Elemental nanoselenium has gained a lot of attention recently since Se nanoparticles (SeNPs) have been reported to exert higher bioavailability, higher biological activity and lower toxicity compared to organic and inorganic Se forms [4,5]. Therefore, they are considered a promising material for many applications, particularly in the field of nutraceuticals and biomedicine [6,7]. SeNPs can be synthesized from inorganic precursors using physical, chemical, or biological synthesis approaches (green synthesis). Physical and chemical processes are often energy-demanding and require the use of toxic chemicals that produce environmental hazards resulting in several application-based limitations in the field of pharmaceuticals/nutraceuticals. Therefore, green syntheses that use natural reducing agents (plants, microorganisms, enzymes, etc.) to change the redox potentials of metals/metalloid-oxyanions and convert them into their nano form are being increasingly investigated. An innovative, quick, simple and cheap approach is using the plant extracts as the source of bioactive compounds for reduction of selenium salts to nanoforms and their subsequent stabilization/surface modifications [3]. Given the increasing demand and numerous areas of possible application of SeNPs, it is necessary to focus on identification and utilization of universally available and sustainable sources of reducing/stabilizing compounds to be used in the novel synthesis processes.

Organic agricultural wastes (OAW) represent an underutilized but universally available rich source of various bioactive compounds (phenolic compounds, terpenes, glucosinolates, dietary fiber, saponins, pigments, etc.) that exert a wide range of biological activities and have established biomedical applications. Considering the current acute climate, ecological crises and constant growth of population, global imperatives are becoming the adoption and successful application of different principles of circular economy (including the reuse and revalorization of OAW) [8]. In the last decade, the applicability of utilizing OAW in the phytofabrication of nanoparticles has been investigated for the synthesis of Fe-, Ag-, Au-, Pt- and Pd-nanoparticles [9,10].

The main goal of this work was to investigate the possibilities of utilization of OAW-derived compounds for obtaining SeNPs with satisfactory physicochemical characteristics. Raw- and purified pectin fractions extracted from mandarin peel were used as stabilizing agents, and polyphenol-rich olive pomace extract (OPE) was used for selenite reduction and SeNP surface modification. Raw materials were chosen primarily based on the availability of mandarin peel and olive pomace in Croatia as part of general attempts to improve OAW reuse and management. Besides, commercially available pectin has already been successfully applied as stabilizing agent in the formulation of metal nanoparticles, including one work that focused on SeNP [11,12,13,14,15]. Recent application of OPE in the synthesis of polyvinylpyrrolidone- and polysorbate-stabilized SeNPs resulted in improved physicochemical characteristics and higher gastrointestinal bioaccessibility [16]. Results obtained within this research will contribute to the current emerging field of green approaches in nanoparticle synthesis, particularly in terms of the reuse of OAW as the source of bioactive components to be utilized in innovative nutraceutical formulation.

## 2. Materials and Methods

### 2.1. Preparation and Chemical Characterization of OPE and Pectin Fractions from OAW

The production and chemical composition of OPE has been described in detail in previous publications [16,17]. Briefly, dried, milled and defatted olive pomace (8 g) was extracted with 60% (*v*/*v*) ethanol-water mixture (400 mL) in a shaking water bath at 70 °C for 2 h. The obtained extract was freeze-dried, ground into a fine powder and stored at −20 °C.

For pectin extraction, a mixture of local Satsuma mandarin peel (*Citrus unshiu Marc.*) and peel from store-bought mandarins was dried at 50 °C for 48 h in an oven dryer (Inko, Zagreb, Croatia), milled and sieved to a particle size of 0.8 mm. The commercial pectin used as the reference was Pectin E440 (Esarom, Austria). Extraction of pectin from mandarin peel was conducted as described by Casas–Orozco and co-workers [18] with some modifications. Ten g of defatted mandarin peel was extracted with 200 mL of 1% citric acid monohydrate (pH 1.5) for 2 h at a temperature of 85 °C, and the reaction mixture was filtered while hot through cotton gauze and filter paper. To precipitate interferents that could affect the purity of the pectin extract, the filtered samples were placed in a refrigerator at +4 °C for 24 h and filtered once again. Pectin precipitation was done by adding twice the amount of 96% ethanol relative to the amount of citric acid. The filtrate and ethanol solution was mixed for 2 h on a magnetic stirrer and placed in the refrigerator overnight to allow complete pectin precipitation. Raw pectin fractions were obtained by simple filtration, followed by drying at room temperature, grinding and storing at 4 °C. Part of raw pectin fractions was additionally washed with 63% (*v*/*v*) ethanol four times to remove the remaining soluble impurities and to obtain purified pectin fraction. Obtained pectin fractions were characterized regarding their equivalent mass (EM), methoxyl content (MC), degree of esterification (DE) and galacturonic acid content (GLA) according to standard analytical procedures [19,20,21].

### 2.2. Synthesis of SeNPs

Prior to SeNP synthesis, lyophilized OPE was dissolved in ultrapure water to give a 10 mg/mL solution and filtered through a 0.45 μm polyethersulfone (PES) syringe filters (Macherey-Nagel, Düren, Germany).

To synthesize SeNPs, 15 mg of raw or purified mandarin peel pectin was weighed directly into a 50 mL Erlenmeyer flask, 28 mL ultrapure water (23 mL for functionalized samples) was added under magnetic stirring (350 rpm), followed by the addition of 1 mL of ascorbic acid (1 M)—acting as a reducing agent and 5 mL of OPE (1%) to the functionalized samples. Finally, 1 mL of Na_2_SeO_3_ (0.1 M) was added dropwise, changing the color of the reaction mixture to red. When the reaction was completed (after 20 min), the reaction mixture was purified from the remaining reactants by dialysis. The end of cellulose dialysis tubing (D9527-100FT, Sigma-Aldrich, St. Luis, MO, USA) was folded twice, closed with a sealing clip (Bevara 6 cm, IKEA, Delft, Netherlands), filled with the reaction mixture, closed and submerged in a beaker filled with 950 mL ultrapure water. The dialysis was performed according to a previously described procedure [16]. The composition of non-functionalized SeNPs stabilized with raw pectin (M) and purified pectin (Mpr) in comparison to OPE-functionalized samples stabilized with raw pectin (Mf) and purified pectin (Mprf) are presented in Table 1.

### 2.3. Physicochemical Characterization and Stability of SeNPs

The measurements of hydrodynamic diameter (dH) and zeta potential (ζ) of the SeNPs were conducted at 25 °C by dynamic light scattering (DLS) and electrophoretic light scattering (ELS) respectively, by using a Zetasizer Ultra (Malvern Instruments, Malvern, UK). Data analysis was carried out using Zetasizer software 2.2 (Malvern Instruments, Malvern, UK). The size distributions and ζ of the samples were measured in a standard disposable cuvette (DTS0012) and disposable folded capillary cell (DTS1070), respectively, after diluting the samples in ultrapure water if needed, and are reported as an average value of 3 measurements.

For the stability measurement, SeNP suspensions were stored at 4 °C in the dark for 30 days. Measurements of hydrodynamic diameter and zeta potential were conducted on the 2nd, 5th, 9th, 20th and 30th days of storage. For measuring the pH value of the samples, a pH meter (SevenMulti, Mettler Toledo, Schwerzenbach, Switzerland) was used.

### 2.4. Measurement of SeNP Antioxidant Activity by Chemical Methods

Antioxidant activity was assessed by the measurement of total reducing capacity (TRC) by the Folin–Ciocalteu method (FC) [22] and Trolox radical scavenging activity (TEAC) by the colorimetric assay originally described by Re and co-authors (1999) [23].

For the TRC measurement, 20 µL of SeNPs (or ultrapure water as blank) was mixed with 50 µL 10% water solution of FC reagent (Sigma-Aldrich, St. Louis, MO, USA) in a 96-well plate. After 5 min of incubation, 160 µL of 0.7 M sodium carbonate was added to each well, and the plate was incubated for 30 min at 37 °C. The absorbance was measured at 750 nm using a Victor X3 plate reader (Perkin Elmer, Waltham, MA, USA). A range of concentrations of gallic acid was prepared to obtain a gallic acid standard curve (3–100 mg/L), and the results are expressed as gallic acid equivalents per mmol of selenium (mg GAE/mmol Se).

For the measurement of TEAC against the 2,2′-azino-bis(3-ethylbenzothiazoline-6-sulfonic) acid radical cation (ABTS^+^), a radical solution was prepared by mixing equal volumes of 7 mmol/L of ABTS and 2.45 mmol/L of potassium persulfate solution. After the 12-h incubation in the dark, the ABTS^+·^ solution was diluted to give an absorbance of 0.70 ± 0.02 at 750 nm. The reaction was conducted in a 96-well plate by mixing 20 μL of the sample/Trolox^®^ standard/blank and 200 μL of adequately diluted ABTS^+·^. The absorbance was measured at 750 nm after 90 s of incubation at 30 °C. The percentage of quenching the absorbance was calculated according to Equation (1):

∆A = (A_0min_ − A_3min_)/A_0min_ × 100
(1)



A calibration curve was generated by plotting different Trolox^®^ concentrations against their respective absorbance-quenching percentages. The antiradical efficiency was expressed as mg of Trolox^®^ equivalents per mmol of selenium (mg TE/mmol Se).

### 2.5. Cell Cultures

Human colorectal adenocarcinoma (Caco-2) and human hepatocellular carcinoma (HepG2) cells were used for SeNP biocompatibility/antioxidant activity investigation. Caco-2 cells were cultured in Dulbecco’s Modified Eagle’s Medium (DMEM; Sigma-Aldrich, St. Louis, MO, USA) supplemented with 20% fetal bovine serum (FBS; Capricorn Scientific, Ebsdorfergrund, Germany), 1% antibiotic/antimycotic (A/A; Sigma-Aldrich, St. Louis, MO, USA), 1% non-essential amino acid (NEAA; Capricorn Scientific, Ebsdorfergrund, Germany) and 4 mM L-glutamine (Sigma-Aldrich, St. Louis, MO, USA). The cells were detached from the flask surface using 1x trypsin (2.5% in HBSS w/o Ca, Mg; Lonza, Basel, Switzerland) diluted in ethylenediaminetetraacetic acid (EDTA; E8008 Sigma-Aldrich, St. Louis, MO, USA) solution and seeded in 96 well plates in a concentration of 20,000 cells per well. The number of cells was estimated using a hemocytometer (Neubauer, Germany). After the seeding, the cells were incubated for 48 h in a CO_2_ incubator (37 °C, 5% CO_2_). The HepG2 cells were cultured in Eagle’s Minimum Essential Medium (EMEM; Sigma-Aldrich, St. Louis, MO, USA) supplemented with 10% FBS, 1% 1% A/ and 4 mM L-glutamine.

#### 2.5.1. Measurement of SeNP Biocompatibility

The toxicity of SeNPs in cell lines was investigated by calculating the respective IC_50_ values in the MTT test, and the obtained results were used for the identification of non-toxic concentrations to be used in further investigation of antioxidant efficiency in cell culture models. Fresh medium was added to the wells, followed by the addition of SeNPs. The cells were incubated with SeNPs for 24 h. After that, SeNPs were removed, and cells were washed once with PBS. 3-(4,5-Dimethylthiazol-2-yl)-2,5-diphenyltetrazolium bromide (MTT) reagent (Carbosynth Limited, Compton, UK) was added in a concentration of 0.5 mg/mL (diluted in PBS), and cells were incubated for 3 h at 37 °C, 5% CO_2_. Dimethyl sulfoxide (DMSO) was used to dissolve formazan crystals, and the plates were shaken for 45 min. The absorbance was measured at 530 nm using a Victor X3 plate reader.

Additionally, the potential of SeNPs to initiate reactive oxygen species (ROS) formation was assessed by 2′,7′-dichlorofluorescin diacetate (DCFH-DA) assay and compared to the effects of known prooxidant tert-butyl hydroperoxide (100 µM) (tBOOH; Sigma-Aldrich, St. Louis, MO, USA) [24]. For that purpose, cells were incubated with 25 µM DCFH-DA (Sigma-Aldrich, St. Louis, MO, USA) for 45 min. The excess dye was then removed, and cells were washed with PBS and treated with SeNPs for an additional 3 h. The fluorescence was measured at 485/535 nm.

For the relative quantification of GSH, a monochlorobimane (mBCl) assay was conducted. The assay is based on a measurement of fluorescence that results from a redox reaction between GSH and mBCL reagent [25]. The culture medium was removed, followed by the addition of the fresh medium. Afterward, the cells were incubated with SeNPs/sodium selenite. Negative controls were incubated with an equivalent volume of ultrapure water that was used for SeNPs’ dilution, and positive controls were incubated with 100 µM of tBOOH in the wells. The treatment solutions were removed after 3 h; cells were washed with PBS and incubated with 40 µM mBCl reagent (Sigma-Aldrich, St. Louis, MO, USA) in PBS for 30 min. The fluorescence intensity was measured at 355/460 nm using a Victor X3 plate reader.

#### 2.5.2. Measurement of SeNP Antioxidant Activity in Cell Models

The efficiency of SeNPs against chemically induced oxidative stress was investigated by measuring the viability of the cells previously incubated with SeNPs for 24 h and after exposure to tBOOH (acting as a prooxidant). Protective effects of SeNPs were measured in terms of cell viability, which was assessed by the MTT test, intracellular ROS formation by the DCHF-DA method and intracellular GSH depletion by the mBCl method, as described previously. For the DCHF-DA method, SeNPs were removed from the wells prior to the addition of tBOOH.

### 2.6. Measurement of Se Content

Selenium content was determined by flame atomic absorption spectrometry (FAAS) on Analyst 800 atomic absorption spectrometer (Perkin Elmer Instruments, Norwalk, CT, USA) with deuterium background correction under optimized measurement conditions with hollow cathode lamp (Perkin Elmer Lumina Single Element Hollow Cathode Lamp) and at optimum flame height (air-acetylene). SeNP samples were wet digested in the microwave digestion unit (Ethos Easy, Milestone Systems, Brondby, Denmark) using HNO_3_ and H_2_O_2_ according to the previously described procedure [26].

### 2.7. Data Analysis

Investigations were conducted in duplicates (characterization of pectin) and quadruplicates (other analyses), and the obtained results were expressed as average ± standard deviations. Obtained results were compared by one-way analysis of variance (ANOVA), and in the case of significant differences, a post hoc Tukey’s multiple comparison test was conducted. The differences between group averages were considered statistically significant if *p* < 0.05. Data were processed using GraphPad^®^Prism 6 Software (San Diego, CA, USA) and Microsoft Office Excel (Redmond, Washington, DC, United States).

## 3. Results and Discussion

The main goal of the conducted research was to investigate the possibilities of utilizing added-value products obtained from OAW (OPE and pectins extracted from mandarin peel) in the green synthesis of SeNPs that would be characterized by satisfactory physicochemical characteristics, biocompatibility and improved antioxidant activity achieved through nanoparticle (NP) surface functionalization. Additionally, the investigation will provide novel and original insight into the physicochemical characteristics of mandarin-derived pectins, which have been very scarcely investigated. The organization of conducted experiments is schematically presented in Figure 1.

### 3.1. Physicochemical Characterization of Mandarin Peel-Derived Pectins

For the plant-assisted green synthesis of SeNPs, mandarin peel-derived pectins were used as stabilization agents, while OPE-derived polyphenols were used for achieving SeNP surface modification that might additionally improve the functionality of obtained formulations.

OPE was obtained by solvent extraction, according to the previously described procedure [17]. OPE´s composition is characterized by a high content of polyphenols that varies depending on the chemical composition of the olive pomace, the applied method of extraction and the drying process. In our laboratory, the obtained yields were usually in the following range: total phenols: 4–10 g/100 g dry extract; hydroxytyrosol: 60–100 mg/100 g dry extract; tyrosol: 15–50 mg/100 g; and oleuropein: 2–30 mg/100 g dry extract [27,28].

The chemical characteristics of raw and purified mandarin pectins are presented in Table 2. It is obvious from the presented yield data (12.8%) that dried mandarin peel can be considered a valuable source of pectin, as compared to other citrus sources; notable, classic solvent extraction was applied in this work; the yields could probably be additionally improved by further optimization of extraction method/process [29]. Originally, low EM pectins were obtained (780 g/mol); however, polymerization (that could be visualized as gel formation) occurred during the purification process, resulting in a significant EM increase (2018.8 g/mol). Obtained data are generally consistent with literature indicating low EM values for raw pectin from different citrus sources (orange and grapefruit) ranging from 381−749 g/mol [30,31], where lower values obtained by other authors can be explained by species-differences, lower pH or higher temperatures applied in the extraction process. Both raw and purified pectins can be considered highly methoxylated and, as such, suitable for sugar-containing and acidic products but with limited gelling properties. Obtained pectins were highly esterified, and the degree of esterification was significantly positively affected by the purification process (69.1% in raw pectin and 86.6% in purified pectin). This value is significantly higher compared to DE obtained for orange peel- and grapefruit-derived pectins [30,31,32]. The content of galacturonic acid in analyzed samples was satisfactory (74.8% in raw- and 69.2% in purified pectins). It reflects the purity of pectin and should be at least 65% if pectins are to be used as food additives or pharmaceutical excipients. This is significantly higher compared to the values obtained for grapefruit peel-extracted pectins (60.95%) [30].

### 3.2. Physicochemical Characterization of SeNPs

The green synthesis approach for obtaining NPs is a promising area in nanotechnology because it is focused on the development of eco-friendly processes that result in minimizing or even eliminating the use of toxic and hazardous chemicals. In this work, the focus has also been set on using naturally occurring biodegradable materials obtained from secondary and easily available raw materials (mandarin peel).

To be utilized for biomedical or food applications, NPs need to possess particular functional and structural properties that distinguish them from discrete molecules or bulk materials. Depending on the raw materials used in the synthesis process and the applied preparation method, a large variety of NP types exist that differ significantly according to their characteristics [33]. The major parameters that determine the basic characteristics of NPs are the particle size and zeta potential. Particle size is one of the main determinants of their bioavailability and biodistribution. Surface charge expressed as the zeta potential is a marker of long-term stability and an indicator of surface characteristics and the related adsorption phenomena [34]. The average size of obtained SeNPs (expressed as hydrodynamic diameter) and zeta potential is presented in Table 3. Obtained diameter values ranged from 171.3 nm (Mpr) up to 216.9 nm (Mf)—values were significantly higher in samples obtained with raw pectins and were slightly increased by functionalization with OPE. The size distribution of the nanoparticles is often the key to their specific, desired physical and chemical properties. Smaller NPs have a relatively large surface area as compared to larger ones; this increases the interaction with biological elements and consequently trigger more toxic and adverse effect, particularly those < 100 nm [35]. Therefore, for food application, NPs with diameters > 100 nm would be desirable. On the other hand, the NP size significantly influences the transmembrane permeability and its interaction with mucus, consequently affecting bioavailability. According to available data, NPs with diameters ≤ 200 nm have satisfactory bioavailability; with further diameter increase, bioavailability is significantly decreased. Investigations show that smaller-size NPs (< 200 nm) are easily reaching the firm part of the mucus layer that is resistant to removal by shear. Also, the cellular uptake and uptake pathway of particles are better for NPs < 200 nm [36]. Considering all of the above, the target size of NPs for food/dietary supplement application would be 100–200 nm which makes purified mandarin peel-derived pectins more suitable for the formulation of SeNPs. The polydispersity indexes of the obtained SeNPs ranged from 0.19–0.25, pointing to narrow, uniform particle-size distribution. Functionalization with OPE significantly decreased the polydispersity index of SeNPs stabilized with raw pectin, but it didn’t affect SeNPs stabilized with purified pectin fraction.

As presented in Figure 2A, the average hydrodynamic diameter decreased during storage. The most notable decrease occurred in non-functionalized SeNPs stabilized with raw pectin (M: 84.73 nm decrease in size), and functionalization with OPE contributed to achieving greater stability (Mf: 38.57 nm decrease in size), while the application of purified pectin resulted in the formation of more stable formulations regardless of functionalization (36.03 nm and 38.13 nm decrease in size in Mpr and Mprf respectively).

Zeta-potential values have long served as indicators of stability against aggregation or deposition, where values above ±30 mV were customarily considered moderately stable against aggregation due to the existence of electrostatic repulsive forces sufficient to prevent aggregation. Zeta-potential values of analyzed samples ranged from −22.3 to −23.1 mV and were not significantly affected by the type of pectin utilized in the synthesis process or OPE functionalization. With absolute values lower than 30 mV, they could be considered potentially unstable; however, it is important to understand that stabilization is also to be achieved by steric stabilization [37]. Zeta potential remained relatively unchanged during 30 days of storage (Figure 2B). With changes in the values of zeta potential and hydrodynamic diameter, a visual assessment on the 30th day of the analysis also showed that all the samples became cloudy.

### 3.3. Biocompatibility of SeNPs

The increasing use of SeNPs will inevitably increase environmental exposure in the future. This highlights the importance of biocompatibility assessments that are nowadays mostly conducted by the spectrophotometric quantification of different dyes commonly used in cytotoxicity assays, such as MTT. Additionally, determining the IC_50_ of SeNP is important in the view of determining non-toxic concentrations of NPs to be used in the investigation of biological activity and mechanisms of action in cell-based models (Figure 3A,B).

As presented in Figure 3A,B, all investigated nanoparticles had lower toxicity compared to inorganic selenium, which showed significantly lower IC_50_ values, consistent with available literature data [16]. Obtained results need to be interpreted with some caution since it has been shown that redox-active metals can catalyze the reduction of tetrazolium salts, so formazan can be generated either in the presence of cellular NAD(P)H or perhaps in the presence of sodium selenite [38]. The possible interfering effect of remaining inorganic selenium in SeNP suspensions has been reduced to a minimum by applying previously validated purification of SeNPs by dialysis [16]. Additionally, the size and coating of particular NPs can alter the magnitude of the reaction kinetics, as has been proven recently for silver nanoparticles and has not been investigated in detail in this work [39]. The positive effect of the utilization of purified pectins as stabilization agents was visible in both cell lines–protective effects of OPE functionalization were also visible (except for the case of the Mprf investigated in Caco-2 cells).

Bearing in mind the mentioned limitations of the MTT test, the cytotoxicity of SeNPs was additionally investigated in terms of their ability to induce cellular oxidative stress by measuring their impact on ROS formation and intracellular GSH depletion. As presented in Figure 3, the prooxidant activity of SeNPs at a concentration of 15 mg/L was significantly lower in comparison to tBOOH as a known prooxidant and, in the case of GSH depletion, also in comparison to inorganic selenium form. The inability of inorganic selenium to induce intracellular ROS formation at applied concentrations (Figure 3C,D) is probably the result of the activity of intracellular defensive mechanisms against oxidative stress occurrence (such as GSH depletion, as presented in Figure 3E,F).

Considering ROS formation, all investigated SeNPs showed a similar effect that was not affected by the type of pectin used for stabilization nor OPE functionalization in both cell lines. However, as shown in the HepG2 cell line, the negative impact on intracellular GSH depletion, even though low for all investigated SeNPs, was less pronounced in SeNPs stabilized with purified pectins (Mpr and Mprf, respectively), with no observed effects of OPE-functionalization. Such differences could not be detected in Caco-2 cell lines which are in line with literature data indicating that liver cells are particularly sensitive to the toxicity of pharmacological doses of SeNPs, and generally more sensitive to redox-induced stress in comparison to Caco-2 [40,41].

### 3.4. Antioxidant Activity of SeNPs

The use of SeNPs in the fields of biomedicine and nutrition is in great part based on their antioxidant activity. Therefore, we were primarily focused on the investigation of this aspect of their biological activity, with special emphasis set on the investigation of the type of stabilization agent used and OPE-functionalization. The first methodological approach used was the measuring of SeNPs´ direct reducing/antiradical quenching properties, which has already been used and proved useful by other authors [42,43,44,45]. Figure 4 clearly shows the advantages of OPE-functionalization. Non-functionalized SeNPs (M and Mpr) showed significantly lower reducing activity in comparison to OPE-functionalized samples (Mf and Mprf) (13.4 and 10.7 g GAE/mol Se vs. 28.8 and 26.7 g GAE/mol Se, respectively). Similarly, ABTS radical scavenging activity was significantly increased by OPE-functionalization (11.4 and 8.6 g TE/mol Se in M and Mpr vs. 19.3 and 22.1 g TE/mol Se in Mf and Mprf). Our previous investigation showed that chemically synthesized polyvinylpyrrolidone stabilized SeNPs when properly purified, didn’t possess significant reducing or radical scavenging capacity [16]. Therefore, the antioxidant activity observed in this work has been mediated by pectins and/or OPE used in the SeNP synthesis process. Namely, recent research showed that citrus peel-derived pectins, particularly those of lower molecular weight, possess significant antioxidant activity and may be useful as a potential natural antioxidant in pharmaceutical and cosmetic industries [46]. As shown in Figure 4, the application of purified pectin in the synthesis process resulted in a slightly decreased reducing potential of SeNPs, but it didn’t affect their TEAC radical scavenging capacity. This might be the consequence of the removal of the residues of reductive sugars that might interfere with the determination of the total reductive potential of the samples [47].

Functionalization of SeNP with OPE increased both types of antioxidative activity—FC reducing potential by 115% (M-Mf) and 149.5% (Mpr-Mprf) and TEAC radical scavenging capacity by 67.8% (M-Mf) and 156.9% (Mpr-Mprf). Observed effects can be explained by the surface modification of functionalized SeNPs with OPE-derived polyphenols, whose significant antioxidant potential has been proven previously [48,49].

Even though fast, simple, cheap and reproducible, chemical-based assays for the determination of antioxidant activity are characterized by numerous limitations and should thus be used only as screening tests and for comparative purposes. Methods used in this work are indirect methods based on the reduction of persistent radical (TEAC) or of inorganic oxidizing species (FC) and therefore measure only one (of many) aspects of antioxidant activity and not even under physiological conditions. It is because of that that they shouldn’t be used for predicting antioxidant activity in biological systems.

Cell-based methods for measuring antioxidant activity are conducted under simulated physiological conditions and consider additional parameters such as biocompatibility, transepithelial permeability (bioavailability) and interaction with other cell components (enabling the measure of indirect antioxidant activity). Therefore, they provide better insight into the biological relevance of the particular antioxidant and present optimal compromise as being significantly simpler and cheaper and with no ethical issues in comparison to animal-based studies.

In this work, the antioxidant activity of SeNPs was investigated in HepG2 and Caco-2 cell lines. The advantage of HepG2 cells is that they express many differentiated hepatic functions and are, as such, most often utilized for biocompatibility, toxicity or metabolism studies [49]. Caco-2 cells were chosen considering the limited bioavailability of SeNPs and the possibility of SeNPs exerting local protective effects on enterocytes. For all experiments, cell lines were incubated with SeNPs for 24 h and then treated with the prooxidant (tBOOH) in a concentration sufficient to produce measurable oxidative stress in the untreated cells.

Preincubation of cells with SeNPs protected both, HepG2 and Caco-2 cells from the negative effects of tBOOH, as indicated by viability values presented in Figure 5A,B, regardless of the type of SeNP tested. The most pronounced protective effects were noticed for Mpr (Caco-2) and Mprf (HepG2). On the other hand, none of investigated SeNPs prevented intracellular ROS formation caused by exposure to tBOOH in either of the tested cell lines (Figure 5C,D). It is important to emphasize that the methodological approach of this assay requires the complete removal of the culture medium with SeNPs, prior to tBOOH addition. Therefore, the potential effects are to be completely dependent on the intracellular concentrations of SeNPs, and their absence indicates low transepithelial permeability of investigated SeNPs. A small but statistically significant protective effect of Mf was observed in the Caco-2 cell line, and this might be due to the increased pinocytosis capacity of Caco-2 cells compared to HepG2 cells and, consequently, more significant intracellular SeNP accumulation. Effects on preventing GSH depletion were more pronounced (Figure 5E,F); the most significant protective effects were obtained with OPE-functionalized SeNPs (Mprf), which is consistent with the results obtained from chemical antioxidant assays.

Obtained data justify the utilization of waste-derived bioactive compounds in the synthesis of SeNPs since it contributes significantly to stability, lower toxicity and improved antioxidative activity of SeNPs. Even though literature data on this particular topic is generally scarce, there are several similar investigations available that also prove the usefulness of the phytomediated SeNP synthesis approach for obtaining improved antioxidant activity [43,44,45,50]. However, most available studies are limited to chemical-based assays and are not assessing the problem of limited bioavailability/permeability of SeNPs. Additionally, future studies should primarily focus on the utilization of easily available secondary raw materials that might serve as the sources of bioactive components to be used in the process of phytomediated synthesis as reducing agents, stabilizers and/or surface modifiers.

## 4. Conclusions

Mandarin peel pectins can be combined with OPE in the green synthesis of highly biocompatible SeNPs, less toxic than inorganic selenium form. Utilization of purified pectin fractions in the synthesis process resulted in smaller average diameters and improved stability. Functionalization with OPE significantly increased the direct radical scavenging antioxidant activity of SeNPs, while the effect of functionalization on cell-based mechanisms of antioxidant defense was not clear. All investigated NPs improved HepG2- and Caco-2 cell viability and protected their intracellular GSH concentrations under induced oxidative stress conditions.

## Figures and Tables

**Figure 1 foods-12-01117-f001:**
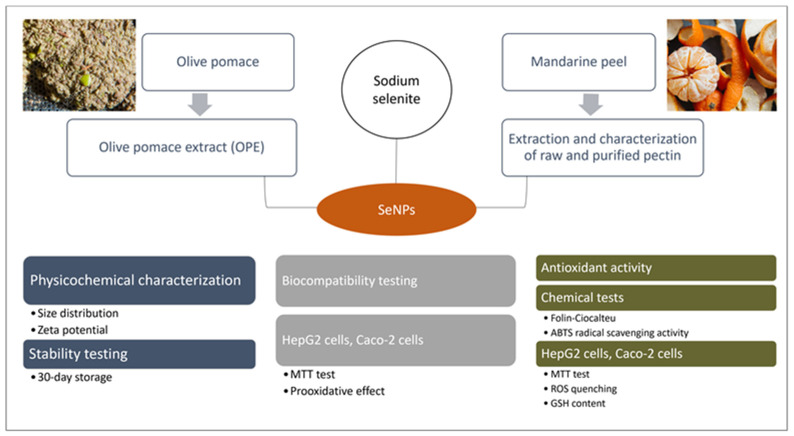
Schematic presentation of the organization of conducted experiments.

**Figure 2 foods-12-01117-f002:**
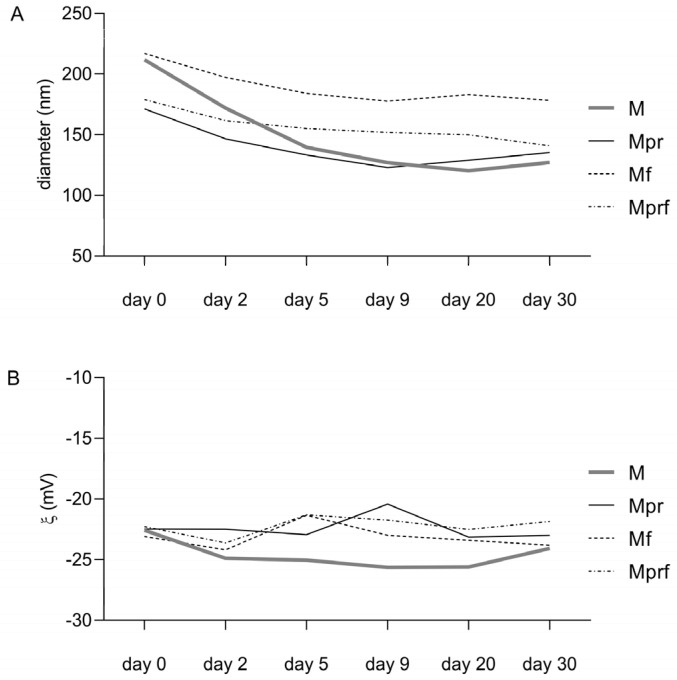
Stability of SeNPs during 30 days of storage regarding average diameter (**A**) and zeta potential (**B**). Data are means of four parallel investigations. M-SeNPs stabilized with raw pectin; Mpr-SeNPs stabilized with purified pectin; Mf-SeNP stabilized with raw pectin and functionalized with OPE; Mprf-SeNPs stabilized with purified pectin and functionalized with OPE.

**Figure 3 foods-12-01117-f003:**
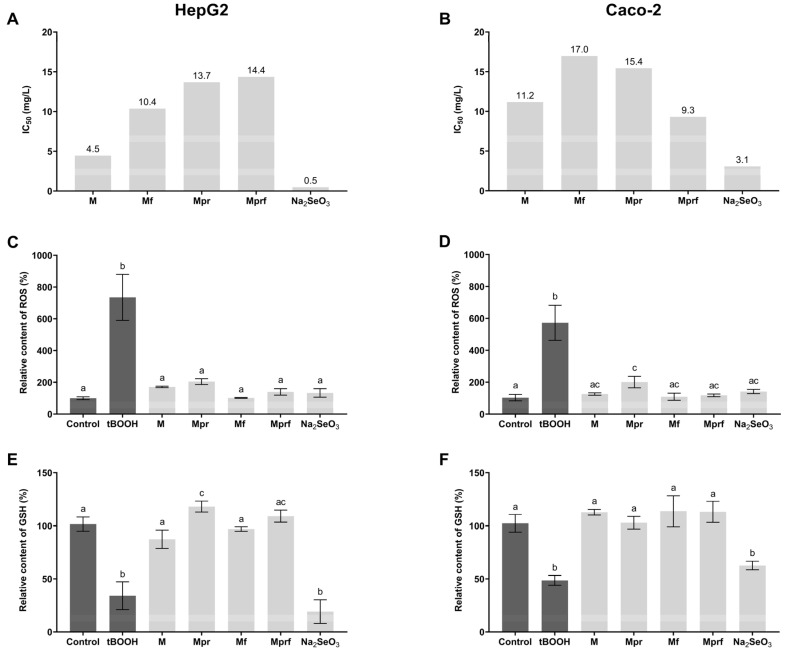
Cytotoxic activity of SeNPs (15 mg/L) in HepG2 (**A**,**C**,**E**) and Caco-2 cells (**B**,**D**,**F**) as assessed by MTT (**A**,**B**), DCFH-DA assay (**C**,**D**) and mBCl assay (**E**,**F**). Data are presented as mean ± standard deviation of four parallel investigations. Differences among investigated samples were analyzed by one-way ANOVA and post hoc Tukey test. Data in the same column, marked with different letters, indicate significant differences (*p* ≤ 0.05). M-SeNPs stabilized with raw pectin; Mpr-SeNPs stabilized with purified pectin; Mf- SeNPs stabilized with raw pectin and functionalized with OPE; Mprf-SeNPs stabilized with purified pectin and functionalized with OPE; control-non treated cells; tBOOH-cells treated with 100 µM tBOOH (500 µM tBOOH for MTT test).

**Figure 4 foods-12-01117-f004:**
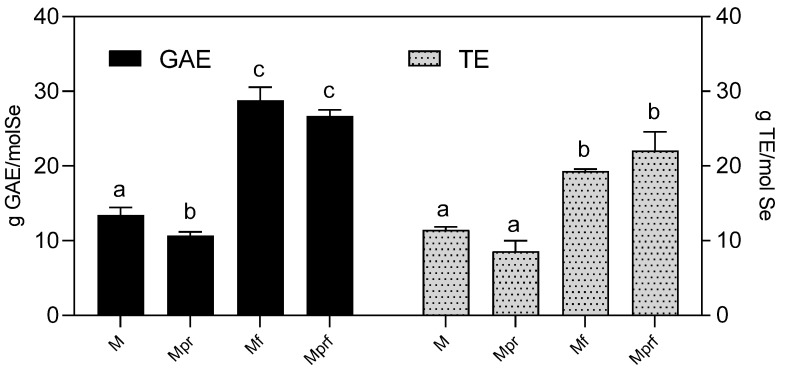
Total reducing potential (expressed as GAE) and radical scavenging activity (expressed as TE) of SeNPs. Data are presented as mean ± standard deviation of four parallel investigations. Differences among investigated samples were analyzed by one-way ANOVA and post hoc Tukey test. Data in the same column, marked with different letters, indicate significant differences (*p* ≤ 0.05). M-SeNPs stabilized with raw pectin; Mpr-SeNPs stabilized with purified pectin; Mf-SeNPs stabilized with raw pectin and functionalized with OPE; Mprf-SeNPs stabilized with purified pectin and functionalized with OPE.

**Figure 5 foods-12-01117-f005:**
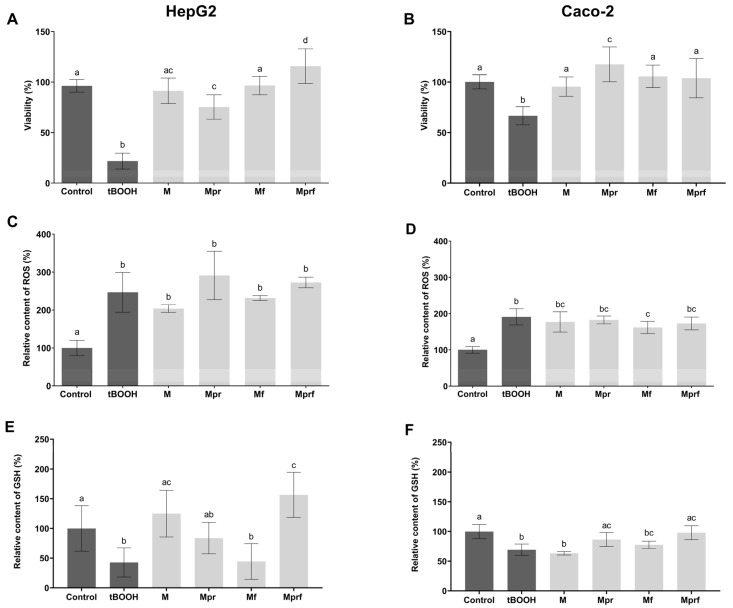
Antioxidant activity of SeNPs (0.1 mg/L) in HepG2 (**A**,**C**,**E**) and Caco-2 cells (**B**,**D**,**F**) as assessed by MTT (**A**,**B**), DCFH-DA assay (**C**,**D**) and mBCl assay (**E**,**F**). Data are presented as mean ± standard deviation of four parallel investigations. Differences among investigated samples were analyzed by one-way ANOVA and post hoc Tukey test. Data in the same column, marked with different letters, indicate significant differences (*p* ≤ 0.05). M-SeNPs stabilized with raw pectin; Mpr-SeNPs stabilized with purified pectin; Mf-SeNPs stabilized with raw pectin and functionalized with OPE; Mprf-SeNPs stabilized with purified pectin and functionalized with OPE; control-non treated cells; tBOOH-cells treated with 100 µM tBOOH.

**Table 1 foods-12-01117-t001:** Composition of reaction mixtures used for the synthesis of SeNPs.

Sample	Na_2_SeO_3_ (0.1 M) (mL)	L-Ascorbic Acid (1 M) (mL)	OPE (1%) (mL)	Raw Pectin (0.05%) (mg)	Purified Pectin (0.05%) (mg)	Ultrapure Water (mL)
M	1	1	0	15	0	28
Mpr	1	1	0	0	15	28
Mf	1	1	5	15	0	23
Mprf	1	1	5	0	15	23

OPE—olive pomace extract; M-SeNPs stabilized with raw pectin; Mpr-SeNPs stabilized with purified pectin; Mf-SeNP stabilized with raw pectin and functionalized with OPE; Mprf-SeNPs stabilized with purified pectin and functionalized with OPE.

**Table 2 foods-12-01117-t002:** Characterization of raw and purified pectin isolated from mandarin peel.

	Yield (%)	EM (g/mol)	MC (%)	DE (%)	GLA (%)
Raw pectin	12.8 ± 0.6 ^a^	780.0 ± 4.1 ^a^	9.1 ± 0.1 ^ab^	69.1 ± 2.4 ^a^	74.8 ± 0.8 ^a^
Purified pectin	7.9 ± 0.5 ^b^	2018.8 ± 10.2 ^b^	10.6 ± 0.6 ^b^	86.6 ± 3.5 ^b^	69.2 ± 0.7 ^b^
Commercial pectin (reference)	/	1954.6 ± 52.6 ^b^	8.7 ± 0.2 ^b^	72.5 ± 1.3 ^a^	59.0 ± 1.2 ^c^

Data are presented as mean ± standard deviation of two parallel investigations. Differences among investigated samples were analyzed by one-way ANOVA and post hoc Tukey test. Data in the same column, marked with different letters, indicate significant differences (*p* ≤ 0.05). EM—equivalent mass; MC—methoxyl content; DE—degree of esterification; GLA—galacturonic acid content.

**Table 3 foods-12-01117-t003:** Average diameter, zeta potential and polydispersity index of SeNPs.

Sample	Average Diameter (nm)	Zeta Potential (mV)	Polydispersity Index	pH
M	211.83 ± 2.47 ^a^	−22.58 ± 0.91 ^a^	0.25 ± 0.00 ^a^	4.02
Mpr	171.33 ± 2.29 ^b^	−22.47 ± 0.90 ^a^	0.20 ± 0.02 ^a^	3.87
Mf	216.87 ± 1.20 ^c^	−23.09 ± 0.83 ^a^	0.19 ± 0.03 ^b^	4.11
Mprf	178.93 ± 1.03 ^d^	−22.27 ± 0.70 ^a^	0.20 ± 0.02 ^a^	3.98

Data are presented as mean ± standard deviation of four parallel investigations. Differences among investigated samples were analyzed by one-way ANOVA and post hoc Tukey test. Data in the same column, marked with different letters, indicate significant differences (*p* ≤ 0.05). OPE—olive pomace extract; M-SeNPs stabilized with raw pectin; Mpr-SeNPs stabilized with purified pectin; Mf-SeNP stabilized with raw pectin and functionalized with OPE; Mprf-SeNPs stabilized with purified pectin and functionalized with OPE.

## Data Availability

Data is contained within the article.

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
