# Peer review of "Phyto-Assisted Synthesis of Nanoselenium–Surface Modification and Stabilization by Polyphenols and Pectins Derived from Agricultural Wastes"

_foods, 2023, doi:10.3390/foods12051117_

Round 1
Reviewer 1 Report
Golub et al studied on phyto-assisted synthesis of nanoselenium – surface modification and stabilization by polyphenols and pectins derived from agricultural wastes. I think this study is meaningful and design of experiments was also logic and novelty. However, some critical issues need to be clarified as follows:
1. The reasons of selected HepG2 and Caco-2 cell as the models used for the assessment of biocompatibility while antioxidant activity was investigated by the combination of chemical and cellular-based assays should be explained.
2. Please explain the reason of selected extracting temperature of 70 °C, if this temperature is good enough for polyphenol activities, why is the freeze-dried selected for drying process? Why not select high temperature hot air drying processing?
3. I’d like to know the yield of polyphenols in OPE from olive pomace.
4. Please explain the reason of methoxy content in purified pectin is a little higher than that in raw pectin from mandarin peel?
5. In the part of 2.2, how to separate the synthesized particles from the reaction solusion?
6. I didn’t find the figure 3 and figure 5 in the pdf file.
7. Conclusion should be concise.
Author Response
Golub et al studied on phyto-assisted synthesis of nanoselenium – surface modification and stabilization by polyphenols and pectins derived from agricultural wastes. I think this study is meaningful and design of experiments was also logic and novelty. However, some critical issues need to be clarified as follows:
DVC: The authors would like to thank the reviewer for their valuable comments. The authors corrected and complemented the manuscript according to the suggestions and hopefully clarified all potential ambiguities. Interventions in the manuscript are marked with green colour.
- The reasons of selected HepG2 and Caco-2 cell as the models used for the assessment of biocompatibility while antioxidant activity was investigated by the combination of chemical and cellular-based assays should be explained.
DVC: I am not sure if You are asking about the use of additional chemical tests in the investigations of antioxidant activity, or You would like us to explain why we chose those particular cell lines for our investigations?
To answer the first question, chemical tests for determination of antioxidant activity used in this work are widely applied because they are fast, inexpensive, highly reproducible and they correlate well with the amounts of polyphenols and other antioxidant substances in investigated extracts/solutions. Because of that, results obtained by this methodology were used as valuable output data during different phases of the optimization of SeNP formulation. Additionally, results obtained by chemical tests were valuable contribution to our understanding of the mechanism of antioxidant action of tested substances that were later observed in cell models (please see Lines 450-465 - antioxidative effects in cell lines were only significant in case SeNPs were in direct contact with ROS and this can be explained by their direct antiradical properties previously proved by chemical tests; effects that required high intracellular concentrations of SeNPs were less prominent, probably due to limited permeability. This is important guide mark for directing future investigations.
For the second question - the reason Caco-2 and HepG2 cells were chosen for the investigations have already been explained in the text of the original manuscript (Lines 442-446)
- Please explain the reason of selected extracting temperature of 70 °C, if this temperature is good enough for polyphenol activities, why is the freeze-dried selected for drying process? Why not select high temperature hot air drying processing? I’d like to know the yield of polyphenols in OPE from olive pomace.
DVC: Obtaining polyphenol rich olive pomace extract (OPE) from olive pomace by classical solvent extraction was optimized during our previous investigation – 120 min of extraction for 2h at 70C resulted with highest yields of total polyphenols and the highest antioxidant activity. (https://hrcak.srce.hr/182916). Regarding the drying process, two types of processes were previously optimized for obtaining dry OPE: spray drying and freeze drying were successfully applied in our laboratory. Hot air drying process was not applicable because of unsatisfactory technological properties of dry extracts obtained in such way.
The following text has been inserted into manuscript, with additional data on OPE composition, including the data about the total phenol content (Lines 252-257): “OPE was obtained by solvent extraction, according to previously described pro-cedure [17]. OPE´s composition is characterized by high content of polyphenols that varies depending on the chemical composition of the olive pomace, applied method of extraction and drying process. In our laboratory obtained yields were usually in the following range: total phenols: 4-10 g/100g dry extract; hydroxytyrosol: 60-100 mg/100g dry extract; tyrosol: 15-50 mg/100g; and oleuropein: 2-30 mg/100g dry extract [27,28].”
- Please explain the reason of methoxy content in purified pectin is a little higher than that in raw pectin from mandarin peel?
DVC: Thank You very much for noticing this. These observations were in consistence with available literature data. However, we re-checked the data and statistical analysis showed that observed differences were not statistically significant after all (p=0.07). In order to avoid such mistakes in interpretation and improve the clarity of presented data, results of the conducted statistical analysis (one way ANOVA and post hoc Tukey test) for all investigated parameters were included in Table 2. Also, the legend under the table has been modified in order to provide all necessary information. The sentence pointing out the significance of the increase of the methoxy content has been removed from the manuscript.
- In the part of 2.2, how to separate the synthesized particles from the reaction solusion?
DVC: This has already been explained in the original version of the manuscript (lines 120-125) and original reference has been provided (describing the optimization of purification process in details)
- I didn’t find the figure 3 and figure 5 in the pdf file.
DVC: I sincerely apologize, the Figures were present in the original (word) version of the manuscript but did not appear in the pdf version. Figure3 and Figure 5 have now been inserted into the manuscript.
- Conclusion should be concise
DVC: Thank You for this suggestion. It has been shortened as suggested (Lines 496-503)

Reviewer 2 Report
Overall, the manuscript deals with interesting topics. However, the manuscript is difficult to evaluate fully because two figures are missing from the manuscript, namely numbers 3 and 5.
Moreover, the methodology needs to be supplemented. There is no information about the number of repetitions of individual experiments. Were the research methods used validated? Are the markings repeatable?The results presented in tables 2 and 3 are averages of how many measurements? What statistical program did the authors of the manuscript use?
Author Response
Overall, the manuscript deals with interesting topics.
DVC: The authors would like to thank the reviewer for their valuable comments. The authors corrected and complemented the manuscript according to the suggestions and hopefully clarified all potential ambiguities. I hope You don’t mind that I grouped Your questions while answering. Interventions in the manuscript are marked with green colour.
However, the manuscript is difficult to evaluate fully because two figures are missing from the manuscript, namely numbers 3 and 5.
DVC: I sincerely apologize, the Figures were present in the original (word) version of the manuscript but did not appear in the pdf version. Figure3 and Figure 5 have now been inserted into the manuscript.
Moreover, the methodology needs to be supplemented. There is no information about the number of repetitions of individual experiments. What statistical program did the authors of the manuscript use? The results presented in tables 2 and 3 are averages of how many measurements?
DVC: The requested data (number of repetitions, statistical analysis of obtained data and statistical programmes used have already been stated in the original version of the manuscript, in the section 2.7. Data analysis (Lines 229-235). You probably missed it by chance. However, to improve the clarity of presented results, Figure- and Table- legends have now been supplemented with requested data (number of repetitions and statistical analysis).
Were the research methods used validated? Are the markings repeatable?
DVC: Yes, for this investigation we used standard, validated methods that are characterized with satisfactory repeatability and accuracy. Original references for all applied methods were stated in the manuscript.

Round 2
Reviewer 1 Report
All issues were addressed.
Reviewer 2 Report
The authors have revised the manuscript in accordance with the review comments. Therefore, the manuscript can be published in its present form.